# Intolerance of uncertainty and mental health in China "Post-pandemic" age: The mediating role of difficulties in emotion regulation

Zi-Hao Gao[1], Jun Li[1,2]*

**1** Department of Education Management, Chinese International College, Dhurakij Pundit University, Bangkok, Thailand, **2** School of Design, Hainan Vocational University of Science and Technology, Haikou, China

* lijun.edu.ma@foxmail.com

**Data Availability Statement:** The data used in this paper are openly available at the OSF site for this paper at https://osf.io/fme5s/, DOI 10.17605/OSF.IO/FME5S.

## Abstract

The Chinese government adjusted its national epidemic prevention and control policy in December 2022 after the worldwide declaration of COVID-19 as a common influenza. After the policy adjustment, there has been widespread infection in China, which has brought a lot of uncertainty to the lives and studies of Chinese university students. This study focused on the impact of the intolerance of uncertainty for COVID-19 (IUC) on the emotional and mental health of college students in China "Post-pandemic" age. This study examined the mediating role of difficulties in emotion regulation (DER) between IUC and mental health (MH). 1,281 university students in China were surveyed using the intolerance of uncertainty for COVID-19 scale, the difficulties in emotion regulation scale and the mental health scale. A structural equation model was used to test the hypothesis model, and it was shown that IUC had a significant negative effect on the MH of college students and a significant positive effect on the DER. DER had a significant negative effect on the MH, and DER had a complete mediation effect between IUC and MH. The findings of this study enrich our understanding of the influencing factors of mental health of university students under the background of post-epidemic in China, and provide practical reference for universities on how to prevent mental health problems under the current uncertain environment in China.

## Introduction

Since the outbreak of COVID-19, the virus has made a major impact on almost every area of human life [1], Governments around the world [2] and many scholars are committed to mitigating the harm caused by the epidemic on human society [3, 4]. With the joint efforts of all countries, the harm of the current epidemic of new coronavirus (Omicron) to the human body has begun to decline [5], which is a view supported by recent research [6]. As the devastating effects of the new coronavirus began to subside, many countries, including China, shifted their focus from strict prevention and control to socioeconomic recovery [7]. In December 2022, the Chinese government designated COVID-19 as seasonal influenza or upper respiratory infection as well as modified and eased its control policy and measures [8].

**Funding:** The author(s) received no specific funding for this work.

**Competing interests:** The authors have declared that no competing interests exist.

China has a high vaccination rate against COVID-19 (approximately 90%), with low levels of infection, severe disease, and mortality rates [9]. However, following the control policy shift in late 2022, there were sudden spikes in COVID-19 infections [10], severe disease cases, and even deaths [5] in China, which might cause panic.

Numerous researchers have noted that there are still uncertainties concerning the rapid transmission and immune evasion of neo-coronaviruses [5, 11–13]. Current research indicates that the increase in the number of COVID-19 infections, the ongoing mutation of the virus, the global economic downturn [14], and the challenging employment situation for graduates [15] have left college students with a great deal of uncertainty regarding their physical wellbeing, learning ways, and future lifestyles [16–18]. Recent research has confirmed that the impact of COVID-19 uncertainty on adolescent psychological symptoms and mental disorders is substantial and persistent [18–20]. It is also proved that COVID-19-related events exacerbate adolescents' intolerance of uncertainty [18]. Intolerance of uncertainty (IU) has been identified as a key predictor of difficulties in emotion regulation (DER) among adolescents during an epidemic [21, 22]. In addition, previous research has noted DER was a significant risk factor for individual mental health (MH) [23, 24]. Therefore, based on the theoretical guidance of the Process Model of Emotion Regulation [25], there are four main research questions in this study: to explore whether the intolerance of uncertainty for COVID-19 (IUC) has a negative effect on the MH of college students; to explore whether the IUC of college students has a positive effect on their DER; to explore whether the DER of college students has a negative effect on their MH; to explore the mediating role of DER between the IUC and MH of college students. The exploration of these questions is helpful to deepen our understanding of the internal influencing factors of college students' MH and also provide empirical evidence and suggestions for educators to guide college students to accurately understand the IU, identify the DER, and then enhance their mental health.

## Theory and research hypotheses

### Process model of emotion regulation

Emotion regulation theory suggests that people can maintain, reduce, or increase individually related emotions or try to develop specific emotions through regulation [25, 26]. Emotion regulation can be a conscious or unconscious process [27, 28]. The process model of emotion regulation divides the process of emotion production and regulation into five continuous and interactive categories based on the different points on an emotion generation timeline [28]. The first four categories, including situation selection, situation modification, attentional deployment, and cognitive change are referred to as "antecedent-focused" because this part focuses on the cognition and modification of the factors that affect the emergence of emotions before they are produced; the last category is termed "response-focused" because it involves modulating an emotion after it has been produced [28, 29]. However, it does not mean that 'antecedent-focused processes' occur in the absence of emotions, and the order in which emotional responses are generated is variable [28, 30, 31]. Within the continuum of the emotion generation process, which consists of successive, overlapping cycles of emotion generation, all emotion regulation processes serve as both a response to the current emotion and a harbinger of upcoming emotions [28, 30].

This process model has contributed significantly in previous studies to explain MH or psychological problems in individuals; good emotion regulation is found essential for social adjustment and overall health, while DER is associated with psychological problems [24, 31–34]. Hu et al. [31] illustrated the relationship between emotion regulation and psychological well-being by meta-analysis, suggesting that effective emotion regulation significantly

impacted an individual's psychological well-being via a series of cognitive processes generated by emotions. De France & Hollenstein [33] demonstrated that emotion regulation strategies were significant predictors of adolescents' MH. A recent latent class analysis identified emotion dysregulation (or DER) as a potential risk factor for psychological symptoms [24].

According to the definition of intolerance of uncertainty, IU is defined as an individual's excessive perception of an uncertain adverse event, circumstance, or situation [35, 36]; individuals with high IU overestimate the likelihood of an adverse event occurring and believe their inability to cope may result in negative emotions [37]. This study examined how IUC influenced the MH of university students via DER by considering IUC as an antecedent-focused factor that might induce DER (an emotionally regulated response) and by taking MH as a psychological state to which DER may contribute.

## IU and MH

As an excessive perception of uncertain circumstances [36], IU has been widely recognized as a significant factor in maintaining psychological problems [35]. Individuals with a high IU have difficulty accepting uncertain events, regardless of how unlikely they are [35]. Previous research has pointed out that IU is strongly associated with mental problems such as general distress [38], anxiety [39, 40], psychological distress [23, 41].

Several recent studies have demonstrated that IU is a significant risk factor and antecedent variable for MH during the COVID-19 pandemic [42–44]. A follow-up study revealed that college students demonstrated a significant increase in psychological symptoms of anxiety and depression in two waves of the survey during the COVID-19 pandemic and that IUC could promote MH issues such as anxiety and depression among college students [19]. The cross-sectional study by Zhuo et al. [45] suggested that COVID-19-related intolerance of uncertainty significantly and negatively predicted MH among returning college students in Wuhan. The results above were also supported by the study by Marín-Chollom and Panjwani [46], which indicated that IU had a detrimental effect on college students' MH during the COVID-19 pandemic. Accordingly, hypothesis 1 is proposed: the IUC of college students significantly negatively affects their MH.

## IU and DER

DER, which is also referred to as emotion dysregulation [47], means that individuals cannot effectively control their negative emotions. Specifically, this means that individuals are unable to correctly understand their emotions, accept their emotional experiences, and control their emotional expression behavior, while at the same time they are unable to flexibly regulate their emotions [48]. Prolonged exposure to harmful situations can deplete a person's ability to regulate emotion, resulting in emotional dysregulation or DER [49].

Grenier et al. [36] noted that individuals with a high IU experience emotional distress when confronted with threats. COVID-19-related threats and IU are strongly associated with negative emotions in college students [50–52]. Previous research has noted that when negative emotions are overwhelming, individuals have difficulty regulating their emotions [49, 53]. A study conducted in the Middle East suggested that IU was associated with DER [23]. Past research has also confirmed that IU is a direct predictor of DER [22, 54]. In a specific study among individuals with missing relatives, chronic grief, and high levels of uncertainty, IU was found significantly and positively predict DER [21]. The study by Zhou et al. [22] conducted in five Chinese cities supported the above results that IU had a significant positive effect on DER. Therefore, Hypothesis 2 is proposed: the IUC of college students significantly positively affects their DER.

### DER and MH

Previous research has suggested that effective emotion regulation is essential for MH [55] and that DER is associated with MH problems [23, 55–57]. Aldao et al. [48] conducted a meta-analysis of 114 past studies. They concluded that good emotion regulation was beneficial for MH and that various types of psychopathology and mental problems could be attributed to DER. A comparative study in India confirmed that DER could contribute to mental health problems and psychiatric issues [58]. The cross-sectional findings of Davoudian et al. [23] also supported the notion that DER was a risk factor for psychological problems and implied that training to improve emotion regulation was beneficial for enhancing MH. Bridges-Curry and Newton [24] identified emotion dysregulation as a significant cause of MH issues. Accordingly, hypothesis 3 is proposed: the DER significantly negatively affects the MH of college students.

### IU, DER and MH

Based on the literature mentioned above, IU may be an essential risk factor for MH [42, 43] and a cognitive variable that triggers DER [22]. Meanwhile, previous research has identified DER as a potential threat to individuals' MH [23, 24].

In addition, DER is frequently reported as a mediating factor in studies on adolescent health issues in various national cultures [57, 59–61]. In particular, Pan et al. [61] discovered a significant mediating effect of DER between the perceived risk of COVID-19 epidemics and internet addiction among Chinese university students. A cross-sectional study of 511 unemployed Chinese youth found that stress had a significant indirect effect on insomnia via the mediating role of emotion dysregulation [59]. Another study among 204 French university students revealed that DER mediated the association between IU and the tendency to worry [60]. DER was also identified as a key mediator in the relationship between attachment and MH disorders among Polish college students [57]. Zhou et al. [22] reported that DER fully mediated the effect of IU on gambling urge and involvement. Hence, hypothesis 4 is proposed: the DER has a mediating role between the IUC and MH of college students.

### The present study

In summary, the recurrence of the COVID-19 epidemic in China, coupled with the revision of the epidemic policy in December 2022, has prompted people to consider whether the virus continues to mutate, how to prevent the epidemic effectively, and the uncertainty of its impact on political and economic life. At the same time, it has aroused our attention to intolerance of uncertainty for the epidemic and the internal influence mechanism of college students' mental health in this context. Inspired by the process model of emotion regulation [25, 27], this study focused on the processes of emotion regulation among college students in China's current context and proposed a mediating model based on previous research to examine how IUC affects college students' MH via DER. This study aimed to extend the previous research and provide practical recommendations on how to help college students understand IU, improve their ability to regulate their emotions, and maintain a healthy mental state in the higher education environment. Accordingly, the hypothetical structural model in this study is depicted in Fig 1.

## Methods and materials

### Research ethic

This study was ethically approved by the authors' institution (Dhurakij Pundit University; No. 037/65). The study also adhered to the Thai National Policy and Guidelines on Human

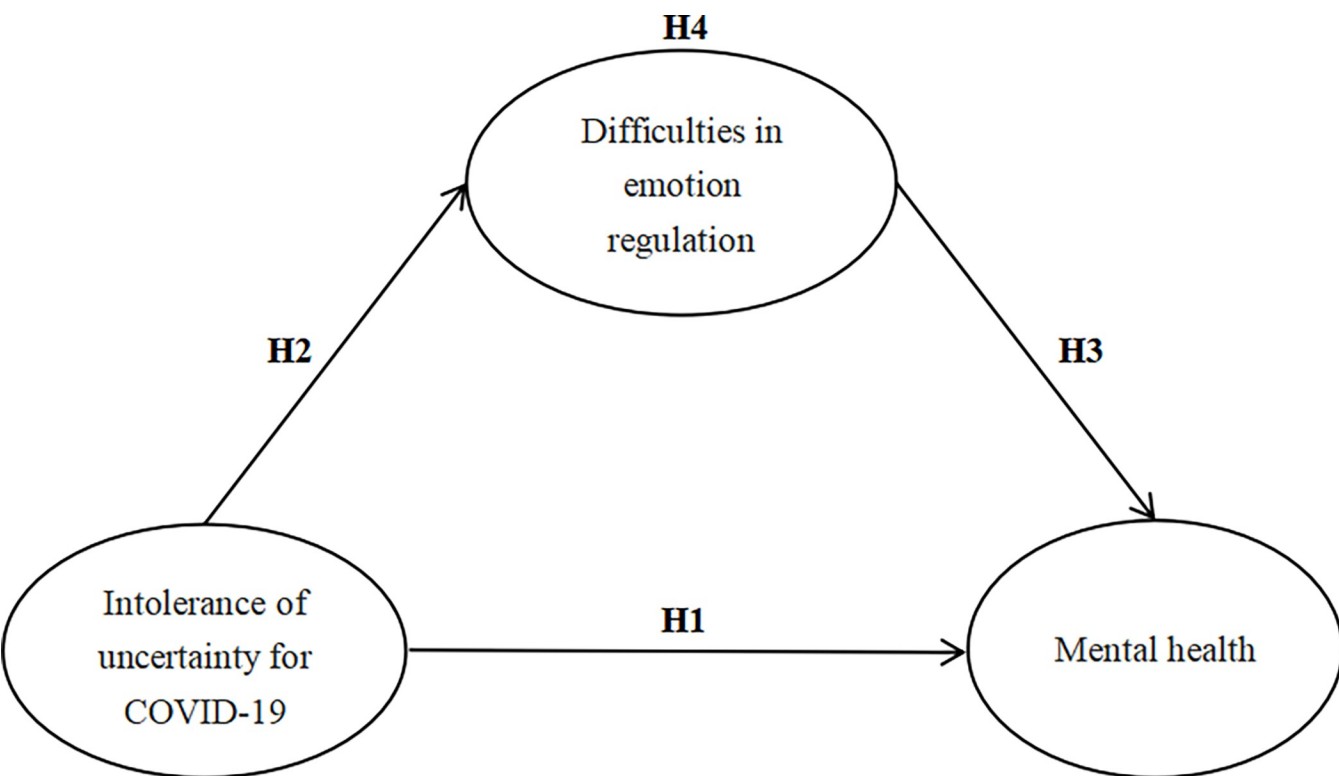

**Fig 1. Hypothetical structural model.** H1: The IUC of college students significantly negatively affects their MH. H2: The IUC of college students significantly positively affects their DER. H3: The DER significantly negatively affects the MH of college students. H4: The DER has a mediating role between the IUC and MH of college students.

Research in respect of participant rights. Participants expressed informed and agreed to participate in this study and gave their written informed consent before data were collected.

## Participants and procedure

As this study was limited by human resources and time, the convenience sampling method was adopted in the data collection procedure, and the survey was carried out at two universities in southern China. The questionnaire was administered online using the Chinese e-questionnaire software Questionnaire Star (www.wjx.cn). Prior to data collection, the authors of this study explained in detail the purpose and the design of the questionnaire to the teachers in charge of distributing questionnaires in the two universities. The corresponding author of this study sent links to the online questionnaire and QR codes to the teachers in charge of the survey, who were available for online supervision. Students at these two universities who were interested in this study topic could voluntarily fill in the questionnaire by scanning the QR code on the poster or clicking on a valid link. This study stated the anonymity, voluntariness and confidentiality of the participants in the preface to the questionnaire. Participants gave informed consent by clicking the Submit button after voluntarily completing the answers. A total of 1,339 college students eventually participated in the survey. The data were collected from March 10 to March 15, 2023 and accessed on March 16, 2023. The corresponding author of this study was responsible for the data extraction and analysis.

1339 questionnaires were collected in this study, and after excluding invalid questionnaires with incomplete answers or short response times, 1281 valid questionnaires remained, with a

valid rate of 95.67%. According to the sampling criteria for formal testing proposed by Ghiselli et al. [62]: if the study involved the use of scales, the number of study samples should not be less than 10 times the total number of items for all the scales. Accordingly, the total number of items in the three scales used in this study is 26, so the effective number of samples in this study meets this sampling criterion.

## Measures

The questionnaire design consisted of two main parts. The first part was a guideline which informed participants of the study's purpose and the instructions for completing the questionnaire. The second part contained 5 questions on demographic characteristics (e.g., what is your gender? Is your university a public/private university?) and three scales.

The Intolerance of Uncertainty for COVID-19 Scale, revised by Luo et al. [63] according to the epidemic context, was employed. It was a unidimensional scale with 4 items (e.g., The uncertainty of COVID-19 has seriously impacted my studies, work, and life). Participants' responses were scored on the 7-point Likert scale ("1–7" represents strongly disagree-strongly agree), a higher score indicated a higher IUC level.

The MH of college students was measured using WHOQOL-Taiwan version, revised by Lin [64]. It was a unidimensional scale with 6 items (e.g., I am satisfied with myself). The last item was a reverse question and was removed from the data analysis. Participants' responses were scored on the 5-point Likert scale ("1–5" represents strongly disagree-strongly agree), a higher score indicated a higher MH level.

The present study measured participants' DER using the Brief Version of the Difficulties in Emotion Regulation Scale (the DERS-16) developed by Bjureberg et al. [65]. The scale consisted of 16 questions divided into 5 dimensions: lack of emotional clarity, difficulties engaging in goal-directed behavior, impulse control difficulties, limited access to effective emotion regulation strategies, nonacceptance of emotional responses (e.g., I have difficulty making sense out of my feelings; when I am upset, I have difficulty focusing on other things.). Responses were scored on the 5-point Likert scale ("1–5" represents almost never-almost always), a higher score indicated a higher DER level.

## Data analyses

SPSS 21.0 and AMOS 22.0 were used to analyze the data. The criteria for statistical significance ($p<0.05$) were adopted for all data analysis processes. SPSS was used for preliminary data analysis, including demographic analysis of the sample, reliability testing of the measurement instruments, and correlation analysis between variables. AMOS was used to test common method variance (CMV) and to test whether the hypotheses of this study were valid. The hypothetical model in this study was validated through a two-step strategy [66]. The first step was to evaluate the measurement model's convergent validity and discriminant validity by confirmatory factor analysis (CFA). The second step was to use the structural equation model (SEM) to construct a mediation model to test its fit and to verify the study hypothesis. Meanwhile, the bootstrapped confidence interval (CI) was used to test the significance of each path coefficient again (sample size was 5000). The 95% CI did not contain 0 to indicate a significant effect [67].

## Results

### The sociodemographic characteristics of participants

Of the valid sample (Table 1), 592 (46.2%) were first-year students, 361 (28.2%) were sophomores, 181 (14.1%) were juniors, 100 (7.8%) were seniors, and 47 (3.7%) were postgraduate

**Table 1. The sociodemographic characteristics of participants.**

| Category | | Number of people | Percentage |
|---|---|---|---|
| Grade | first-year students | 592 | 46.2% |
| | sophomores | 361 | 28.2% |
| | juniors | 181 | 14.1% |
| | seniors | 100 | 7.8% |
| | postgraduate students | 47 | 3.7% |
| Gender | male | 415 | 32.4% |
| | female | 866 | 67.6% |
| Nature of colleges | public undergraduate institutions | 702 | 54.8% |
| | private undergraduate institutions | 579 | 45.2% |

students. 415 of them (32.4%) were male, and 866 (67.6%) were female. There were 702 (54.8%) students from public undergraduate institutions and 579 (45.2%) from private undergraduate institutions. The age range was 18–26 years.

## CMV test

The CMV was evaluated prior to data analysis to demonstrate that findings would not be significantly influenced [68]. This study conducted CFA to include all questions of the 7 potential variables of the three scales and then compared the fit of the single-factor model with the fit of the seven-factors model. The results are shown in Table 2, where the fit indices of the seven-factors model were significantly better than those of the single-factor model ($\triangle x^2$ = 13790.98, $\triangle df$ = 21, $p<0.001$), indicating that the CMV problem in this study was not severe [68, 69].

## Reliability and validity assessment

The consistency of the instruments used in this study was checked. The Cronbach's α was 0.932 of the IUC scale; 0.918 of the MH scale; 0.886, 0.963, 0.965, 0.961, and 0.896 on each dimension of the DERS (Table 3). All were greater than 0.7, indicating that the measurement instruments in this study had good reliability [70].

Moreover, CFA was performed for all three measurement models prior to SEM analysis. The factor loadings were greater than 0.5 for all three measurement models; the values of composite reliability (CR) for each dimension were greater than 0.7; the values of average variance extracted (AVE) were greater than 0.5 (Table 3), the results stated that the convergence validity of the measurement model was satisfactory [71]. The square root of the AVE of each dimension was greater than the correlation coefficient between each dimension (Table 4), indicating the discriminant validity of the measurement model was favorable [72].

## Descriptive statistics and correlation analysis

Table 5 showed that IUC was significantly negatively associated with MH (r = -0.099, $p<0.001$); IUC was significantly positively associated with DER (r = 0.382, $p<0.001$); DER was significantly negatively associated with MH (r = -0.266, $p<0.001$) [71].

**Table 2. Comparison of the model fitness.**

| Model | $X^2$ | DF | $X^2/DF$ | $\triangle X^2$ | $\triangle DF$ | P | RMR | GFI | CFI | NFI |
|---|---|---|---|---|---|---|---|---|---|---|
| Single-factor Model | 15317.75 | 275 | 55.70 | 13790.98 | 21 | P < 0.00001 | 0.278 | 0.463 | 0.571 | 0.567 |
| Seven-factor Model | 1526.77 | 254 | 6.01 | | | | 0.053 | 0.906 | 0.964 | 0.957 |

**Table 3. Reliability and convergence validity of the measurement model.**

| Dimension | Item | Factor Loading | CR | AVE | Cronbach's α |
|---|---|---|---|---|---|
| IUC | 1. The uncertainty of the pandemic has seriously impacted my studies, work and life. | 0.870 | 0.932 | 0.775 | 0.932 |
| | 2. The uncertainty of the pandemic ruins my plans. | 0.874 | | | |
| | 3. The uncertainty of the pandemic makes me uneasy, anxious, or stressed. | 0.887 | | | |
| | 4. Confronted with the uncertainty of the pandemic, I cannot function very well in my studies, work and life. | 0.890 | | | |
| MH | 1. I enjoy my life. | 0.824 | 0.917 | 0.690 | 0.918 |
| | 2. I feel that my life is meaningful. | 0.872 | | | |
| | 3. I can focus on what I want to do, such as thinking, studying, memorizing, and so on. | 0.854 | | | |
| | 4. I can accept my appearance. | 0.788 | | | |
| | 5. I am satisfied with myself. | 0.813 | | | |
| LEC | 1. I have difficulty making sense out of my feelings. | 0.851 | 0.889 | 0.800 | 0.886 |
| | 2. I am confused about how I feel. | 0.936 | | | |
| DEGB | 1. When I am upset, I have difficulty getting work done. | 0.931 | 0.963 | 0.897 | 0.963 |
| | 2. When I am upset, I have difficulty focusing on other things. | 0.960 | | | |
| | 3. When I am upset, I have difficulty thinking about anything else. | 0.950 | | | |
| ICD | 1. When I am upset, I become out of control. | 0.941 | 0.966 | 0.904 | 0.965 |
| | 2. When I am upset, I feel out of control. | 0.965 | | | |
| | 3. When I am upset, I have difficulty controlling my behaviors. | 0.946 | | | |
| LA | 1. When I am upset, I believe that I will remain that way for a long time. | 0.918 | 0.961 | 0.832 | 0.961 |
| | 2. When I am upset, I believe that I will end up feeling very depressed. | 0.929 | | | |
| | 3. When I am upset, I believe that there is nothing I can do to make myself feel better. | 0.923 | | | |
| | 4. When I am upset, I start to feel very bad about myself. | 0.893 | | | |
| | 5. When I am upset, my emotions feel overwhelming. | 0.896 | | | |
| NER | 1. When I am upset, I feel ashamed with myself for feeling that way. | 0.798 | 0.898 | 0.746 | 0.896 |
| | 2. When I am upset, I feel like I am weak. | 0.884 | | | |
| | 3. When I am upset, I become irritated with myself for feeling that way. | 0.906 | | | |

Note: LEC, Lack of Emotional Clarity; DEGB, Difficulties Engaging in Goal-Directed Behavior; ICD, Impulse Control Difficulties; LA, Limited Access to Effective Emotion Regulation Strategies; NER, Nonacceptance of Emotional Responses.

## Structural equation model

First, the total effect of IUC on MH was tested using SEM (Fig 2). The model fit indices for the main effects model were: CMIN = 685.03, DF = 26, GFI = 0.886, CFI = 0.928, NFI = 0.926, IFI = 0.928, TLI = 0.901, PCFI = 0.670. The results showed that IUC significantly and

**Table 4. Discriminant validity.**

| Dimension | M | SD | 1 | 2 | 3 | 4 | 5 | 6 | 7 |
|---|---|---|---|---|---|---|---|---|---|
| IUC | 4.983 | 1.506 | 0.880 | | | | | | |
| MH | 3.456 | 0.880 | -0.099*** | 0.831 | | | | | |
| LEC | 2.518 | 1.032 | 0.317*** | -0.166*** | 0.894 | | | | |
| DEGB | 2.825 | 1.099 | 0.436*** | -0.175*** | 0.620*** | 0.947 | | | |
| ICD | 2.249 | 1.101 | 0.271*** | -0.238*** | 0.639*** | 0.653*** | 0.951 | | |
| LA | 2.365 | 1.036 | 0.344*** | -0.297*** | 0.679*** | 0.697*** | 0.792*** | 0.912 | |
| NER | 2.329 | 0.996 | 0.296*** | -0.234*** | 0.641*** | 0.671*** | 0.729*** | 0.821*** | 0.864 |

Note: n = 1281; The underlined numbers are the square root of AVE; the numbers in the lower diagonal are the correlation coefficients; ***$p<0.001$.

**Table 5. Descriptive statistics and correlation analysis.**

| Variable | *M* | *SD* | IUC | MH | DER |
|---|---|---|---|---|---|
| IUC | 4.983 | 1.506 | 1 | | |
| MH | 3.456 | 0.880 | -0.099*** | 1 | |
| DER | 2.442 | 0.926 | 0.382*** | -0.266*** | 1 |

Note: n = 1281; ***$p<0.001$.

negatively predicted MH ($\beta$ = -0.123, $p<0.001$; 95%CI = -0.192, -0.035), and hypothesis 1 was supported.

Secondly, the second-order mediation model (Fig 3) was constructed using SEM, and the model fit indices were CMIN = 1692.289, DF = 267, GFI = 0.900, CFI = 0.959, NFI = 0.952, IFI = 0.959, TLI = 0.954, PCFI = 0.854. It indicated the excellent fitness of the hypothetical model in this study [73].

As shown in Fig 3, after adding DER as a mediating factor, there was no significant direct effect of IUC on MH($\beta$ = -0.002, $p>0.05$; 95% CI = -0.076, 0.083), while a significant positive effect of IUC on DER was observed ($\beta$ = 0.402, $p<0.001$; 95% CI = 0.344, 0.455), and hypothesis 2 was supported; a significant negative effect of DER on MH could also be observed ($\beta$ = -0.300, $p<0.001$; 95% CI = -0.379, -0.216), and hypothesis 3 was supported. It was shown that DER completely mediated IUC and MH.

In addition, 95% bootstrapped CI was used to again estimate each effect value for the hypothesized model. As shown in Table 6, the indirect effect of IUC on college students' MH through DER was observed with the effect value of -0.121 (95% CI = -0.160, -0.085) while the direct effect was not significant ($\beta$ = -0.002, 95% CI = -0.076, 0.083), indicating a fully mediated model in this study. The total effect value was -0.123 (95% CI = -0.192, -0.035), and the mediated effect accounted for 98.374% of the total effect.

## Discussion

The findings of this study indicate that college students' IUC can significantly and negatively predict their MH. This result is comparable to previous research confirming that IU is associated with psychological problems during the COVID-19 pandemic [23] and can trigger MH

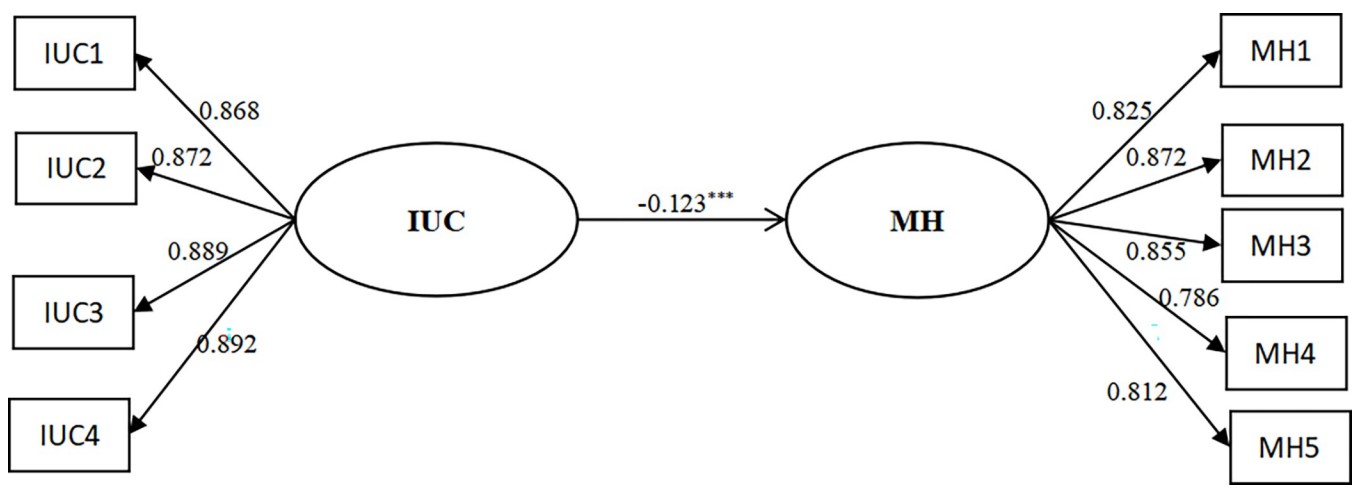

**Fig 2. Main effect model.**

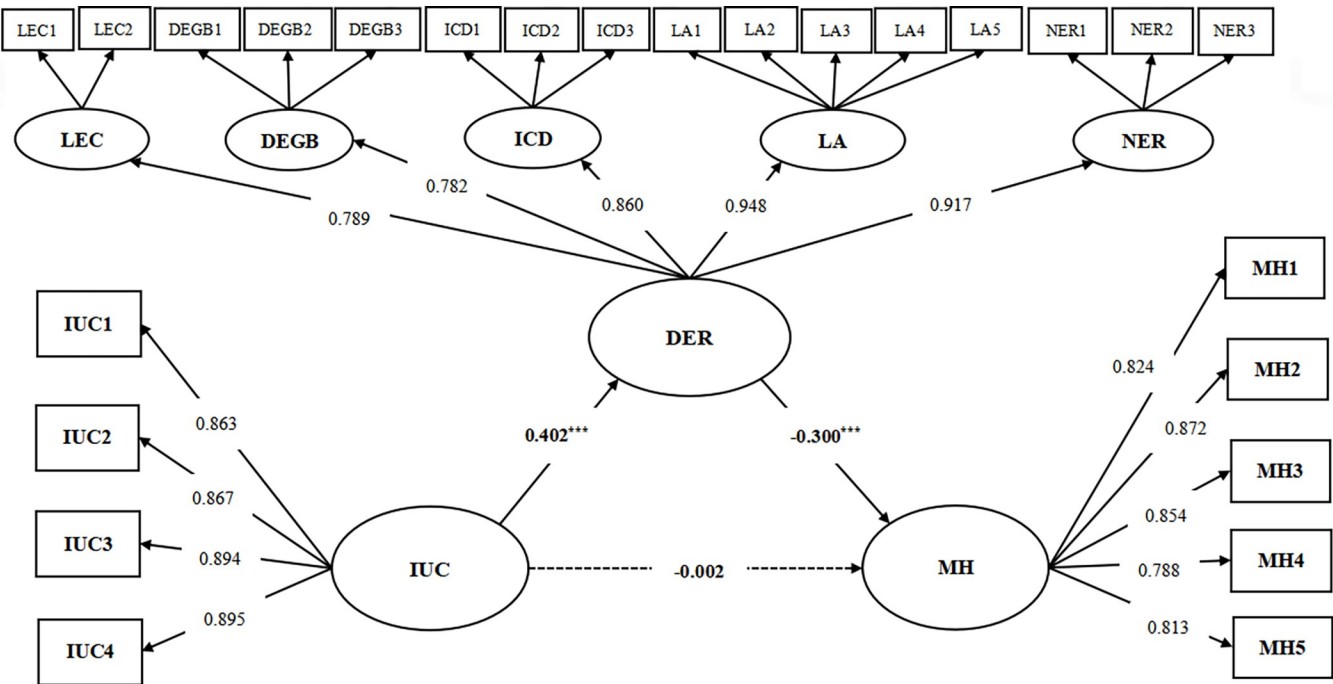

**Fig 3. The second-order mediation model.** Note: IUC, Intolerance of Uncertainty for COVID-19; MH, Mental Health; LEC, Lack of Emotional Clarity; DEGB, Difficulties Engaging in Goal-Directed Behavior; ICD, Impulse Control Difficulties; LA, Limited Access to Effective Emotion Regulation Strategies; NER, Nonacceptance of Emotional Responses.

issues [38–41]. A possible explanation for this result is that the recurrence of the epidemic and widespread infections as a result of the recent adjustment of prevention and control policies in China [5, 10] have caused college students' concern and anxiety regarding the continued mutation of the new coronavirus, their physical health, schooling, and employment in numerous ways. Meanwhile, IUC puts college students in a state of tension and distress and may cause mental health problems [38–40].

Consistent with previous research, this study confirms that IUC significantly and positively predicts DER [22, 54]. The reason may be that there are still large uncertainties in the current epidemic situation [5, 10], which may once again test college students' tolerance to uncertainties such as their learning style, living environment, and life safety, resulting in greater persistent negative emotions [50–52], and such persistent negative emotions will affect individual cognition [74], further amplifies the harm of the epidemic, which makes it difficult for college students to accept although the probability of the epidemic hurting them again is not large, which in short is DER [35].

**Table 6. Bootstrapped confidence interval.**

| Path | | Estimate | 95% Confidence Interval | |
|---|---|---|---|---|
| | | | Lower Limit | Upper Limit |
| Direct effect | IUC→MH | -0.002 | -0.076 | 0.083 |
| | IUC→DER | 0.402 | 0.344 | 0.455 |
| | DER→MH | -0.300 | -0.379 | -0.216 |
| Indirect effect | IUC→DER→MH | -0.121 | -0.160 | -0.085 |
| Total effect | IUC→MH | -0.123 | -0.192 | -0.035 |

This study also verifies that DER significantly negatively predicts MH, which is also consistent with previous research [23, 57]. At the same time, the results of this study extend the above studies and find a mediating model, that is, DER has a mediating effect between IUC and MH of college students. This result is similar to that of Zhou et al. [22], which confirms that DER is a key mediating factor in the impact of UI on MH.

It is worth noting that this mediation model is a completely mediated model, and DER fully mediates the influence of IUC on MH. According to the results of correlation analysis in this study, there is a low correlation between IUC and MH, which indicates that there may be important mediating factors between IUC and MH. The result of this study finds that DER has a complete mediating effect between IUC and MH, which also provides an explanation for this low correlation. Specifically, the uncertainty of the current epidemic may touch the bottom line of college students' tolerance to the uncertainty of their learning style, living environment, life safety, etc., and causes large persistent negative emotions [50–52]. Such persistent negative emotions continue to accumulate in the reality of the current epidemic recurrence and large-scale infection in society, which may affect college students' cognition [74], magnify their understanding of the epidemic harm, and make it difficult to accept even if the probability of uncertain time is not large [35], ultimately lead to common mental health problems among college students [58]. This completely mediated model indicates that DER has a more critical impact on the MH of college students than IUC.

## Contribution

First of all, in terms of theoretical contribution, this study finds that IUC of college students has a significant impact on DER, and DER also has a significant impact on MH, which once again verifies the Process Model of Emotion Regulation in the group of college students. Specifically, within the continuum of the emotion generation process, which consists of successive, overlapping cycles of emotion generation, all emotion regulation processes serve as both a response to the current emotion and a harbinger of upcoming emotions [28, 30]. At the same time, it also supports previous studies and once again confirms that good emotional regulation is important for an individual's social adaptation and overall health, while DER may cause MH problems [24, 31–34].

Secondly, although there are numerous previous studies on the impact of DER and IU on MH respectively, the previous studies considering the common impact of DER and IU on MH are scarce, this study provides empirical evidence for examining the relationship between the process model of emotion regulation and MH [31], a completely mediated model is validated, which fills this gap.

Thirdly, this study finds that if there are repeated outbreaks of global infectious diseases similar to COVID-19, college students may IUC, which may trigger DER and eventually affect MH. This is one of the practical contributions of this study. Fourth, this study also finds that DER has a key impact on MH problems caused by IUC. This provides a practical reference for national education authorities and universities to deal with global infectious diseases in the future.

## Suggestion

A complete mediation model was validated in our study to explain how college students' IUC affects their MH through DER. The results demonstrated that DER completely mediated the effect of IUC on MH. In light of China's recurrent epidemic, this result provides university educators with suggestions for developing effective measures to prevent and improve the MH of college students:

First, in the current state of recurrence and uncertainty of the epidemic in China, college teachers should provide authoritative and scientific information and the most recent research results related to the COVID-19 pandemic in order to improve college students' correct understanding and reduce their intolerance of the epidemic's uncertainty due to misplaced information. They can also share scientific prevention strategies to alleviate college students' fears of infection. In light of the current state of the epidemic's development in China, colleges and universities should make proper teaching and living arrangements for students, actively and effectively assist them in improving their employability, and mitigate the negative impact of the epidemic's uncertainty in every possible aspects.

Second, college teachers should be acutely aware of the emotional and psychological problems caused by students' IUC and provide training or lectures on psychological counseling to assist them in improving their ability to accept uncertainty and self-regulation of emotions, and provide group emotional regulation training to reduce their DER levels. Moreover, effective emotion regulation strategies can be taught to reduce the risk of emotional disorders and preserve MH among college students.

## Conclusion

Focusing on the MH of Chinese college students following the adjustment of the Chinese government's epidemic prevention and control policy, this cross-sectional study found that the IUC of college students had a significant negative effect on their MH under the current state of the recurrent epidemic and widespread infection in China; the IUC of college students had a significant positive effect on their DER; the DER of college students had a significant negative effect on their MH. Furthermore, this study discovered that DER completely mediated the IUC and MH. This study supported the process model of emotion regulation by confirming that UI was an antecedent variable of college students' DER in particular situations and extended the association between this model and MH. The findings highlighted the significance of college students' DER and suggested that educators should pay more attention to this crucial factor while also focusing on college students' IUC. The results of this study provided empirical evidence on how universities can protect against college students' mental health problems.

## Limitations and future research recommendations

First, this study used a cross-sectional design, so the results do not infer causal logic between variables, and future studies may consider adopting a longitudinal studies to further explore the causal associations between these variables. Second, the population of participants was comprised of college students; however, other groups, such as the elderly and the disabled, could be considered in the future to increase the generalizability of the findings. Last but not least, the mediation model constructed in this study was only one model to explain relationship between variables. Because the correlation between IUC and MH was very low, future studies could consider exploring other mediators or moderators to further analyze the strength of the variable relationship to more comprehensively explain the mental health of college students.

## Author Contributions

**Conceptualization:** Zi-Hao Gao, Jun Li.

**Data curation:** Jun Li.

**Formal analysis:** Zi-Hao Gao.

**Investigation:** Jun Li.

**Methodology:** Jun Li.

**Writing – original draft:** Zi-Hao Gao, Jun Li.

**Writing – review & editing:** Zi-Hao Gao, Jun Li.

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
