## [Decision Letter · Decision Letter 0]

4 Jul 2023

PONE-D-23-13232Intolerance of Uncertainty and Mental Health in China “Post-pandemic” Age: The Mediating Role of Difficulties in Emotion RegulationPLOS ONE

Dear Dr. Li,

Thank you for submitting your manuscript to PLOS ONE. Firstly, we would like to apologize for the delay in processing your manuscript. It has been exceptionally difficult to secure reviewers to evaluate your study. We have now received one completed review, which is available below. The reviewer has raised significant scientific concerns about the study that need to be addressed in a revision.

Please note that we have only been able to secure a single reviewer to assess your manuscript. We are issuing a decision on your manuscript at this point to prevent further delays in the evaluation of your manuscript. Please be aware that the editor who handles your revised manuscript might find it necessary to invite additional reviewers to assess this work once the revised manuscript is submitted. However, we will aim to proceed on the basis of this single review if possible. 

After careful consideration, we feel that it has merit but does not fully meet PLOS ONE’s publication criteria as it currently stands. Therefore, we invite you to submit a revised version of the manuscript that addresses the points raised during the review process.

We look forward to receiving your revised manuscript.

Kind regards,

Miquel Vall-llosera Camps

Senior Editor

PLOS ONE

Journal Requirements:

Reviewers' comments:

Reviewer's Responses to Questions

**Comments to the Author**

1. Is the manuscript technically sound, and do the data support the conclusions?

Reviewer #1: Yes

2. Has the statistical analysis been performed appropriately and rigorously? 

Reviewer #1: Yes

3. Have the authors made all data underlying the findings in their manuscript fully available?

Reviewer #1: No

4. Is the manuscript presented in an intelligible fashion and written in standard English?

Reviewer #1: Yes

5. Review Comments to the Author

Reviewer #1: I have several concerns with presented manuscript in its current version mainly due to the methodological aspects.

1. The participants are not described in the introduction . I would recommend adding a summary table of the sociodemographic characteristics of the study participants.

2. The authors should add more details on sample selection. It is not completely clear how was it selected. Additionally, the authors do not propose a minimum sample size, and it is not clear what the reference population is?

3. The authors found a very low correlation of(- .09) between IUC and MH. Although it turned out to be significant, the assumed causal relationship that IUC led to MH seems questionable. The authors should emphasize that in the discussion and underline the need of further research to analyze the strength of these relationships.

6. PLOS authors have the option to publish the peer review history of their article (what does this mean?). If published, this will include your full peer review and any attached files.

Reviewer #1: No

---

## [Author Response · Author response to Decision Letter 0]

18 Aug 2023

Response to Reviewer 1 Comments

Dear reviewer 1:

Thank you very much for your time involved in reviewing the manuscript and your comments have further improved the quality of the manuscript.

We have carefully reviewed the comments and revised the manuscript accordingly. The modified section was already highlighted in yellow. Hope the explanation has fully addressed all of your concerns. Point-by-point response to reviewer are attached below this letter.

Please see the attachment.

Point 1: The participants are not described in the introduction. I would recommend adding a summary table of the sociodemographic characteristics of the study participants. 

Response 1: Many thanks for your comments, we have added a summary table of the sociodemographic characteristics of the study participants in the Results section (Table 1), for specific modifications see lines 246-252 of the manuscript.

Point 2: The authors should add more details on sample selection. It is not completely clear how was it selected. Additionally, the authors do not propose a minimum sample size, and it is not clear what the reference population is? 

Response 2: Thank you for your suggestions. We have added details about the sample selection and the reasons for the selection, see lines 186-201 of the manuscript for specific corrections.

Also, we have supplemented the criteria for minimum sample size, see lines 204-208 of the manuscript for specific modifications.

Point 3: The authors found a very low correlation of(- .09) between IUC and MH. Although it turned out to be significant, the assumed causal relationship that IUC led to MH seems questionable. The authors should emphasize that in the discussion and underline the need of further research to analyze the strength of these relationships.

Response 3: Your advice makes a lot of sense. We speculate that the low correlation between IUC and MH also constitutes a realistic reason for the complete mediating role of DER between IUC and MH in this study. Therefore, we have supplemented and explained this reason in the Discussion section, as detailed in lines 338-342 of the manuscript.

Furthermore, since this study was a cross-sectional design, a causal relationship was not assumed. This is mentioned in the Limitations section (as detailed in lines 382-384). At the same time, we also underlined the need for further research to analyze the strength of these relationships in the Research limitations section, as detailed in lines 388-391 of the manuscript.

---

## [Decision Letter · Decision Letter 1]

13 Nov 2023

PONE-D-23-13232R1Intolerance of Uncertainty and Mental Health in China “Post-pandemic” Age: The Mediating Role of Difficulties in Emotion RegulationPLOS ONE

Dear Dr. Li,

Thank you for submitting your manuscript to PLOS ONE. After careful consideration, we feel that it has merit but does not fully meet PLOS ONE’s publication criteria as it currently stands. Therefore, we invite you to submit a revised version of the manuscript that addresses the points raised during the review process.

ACADEMIC EDITOR: Dear Authors,please revise your manuscript carefully. All the best

We look forward to receiving your revised manuscript.

Kind regards,

Nebojsa Bacanin

Academic Editor

PLOS ONE

Additional Editor Comments (if provided):

Dear Authors,

please revise proposed manuscript thoroughly according to all reviewers' comments.

Additionally, please do the following:

- Visualization of obtained results must be improved

- Motivation behind proposed research should be more clearly explain. Please elaborate what is "beyond state-of-the-art" of proposed. study.

- Make sure that the source code is available according to PLOS ONE publication policies.

- Make sure that you have conducted rigid statistical analysis.

All the best,

AE

Reviewers' comments:

Reviewer's Responses to Questions

**Comments to the Author**

1. If the authors have adequately addressed your comments raised in a previous round of review and you feel that this manuscript is now acceptable for publication, you may indicate that here to bypass the “Comments to the Author” section, enter your conflict of interest statement in the “Confidential to Editor” section, and submit your "Accept" recommendation.

Reviewer #1: All comments have been addressed

Reviewer #2: (No Response)

Reviewer #3: All comments have been addressed

2. Is the manuscript technically sound, and do the data support the conclusions?

Reviewer #1: Yes

Reviewer #2: Partly

Reviewer #3: Yes

3. Has the statistical analysis been performed appropriately and rigorously? 

Reviewer #1: Yes

Reviewer #2: Yes

Reviewer #3: Yes

4. Have the authors made all data underlying the findings in their manuscript fully available?

Reviewer #1: Yes

Reviewer #2: Yes

Reviewer #3: No

5. Is the manuscript presented in an intelligible fashion and written in standard English?

Reviewer #1: Yes

Reviewer #2: Yes

Reviewer #3: Yes

6. Review Comments to the Author

Reviewer #1: The authors have adequately addressed my comments raised in a previous round of review and I feel that this manuscript is now acceptable for publication.

Thank you.

Reviewer #2: 1. State the research question clearly in the Introduction.

2. List the main contributions in the Introduction.

3. Provide paper structure at the end of the Introduction.

4. Literature survey should be expanded, especially with recent papers dealing with COVID19. You can include the following:

https://www.sciencedirect.com/science/article/pii/S2210670720308842

https://link.springer.com/article/10.1007/s00521-022-07424-w

https://link.springer.com/chapter/10.1007/978-981-19-1653-3_35

https://www.sciencedirect.com/science/article/pii/S0045790622003159

https://www.nature.com/articles/s41598-022-06218-3

5. The contribution of the research to the field should be justified better.

6. Provide more details about the utilized questionary.

7. Discussion should be more elaborate.

8. Discuss the limitations of the research as well.

9. Conclusion is very limited - it should summarize up the research, mention the most important findings and limitations of the research, and indicate future research in this topic.

10. Thorough proofreading of the paper is recommended.

Reviewer #3: The authors took care of the recommendations. However, the data set that is made available is not intelligible. The columns of the table are not in English and the meaning of the values is not explained.

Please add a description of the data set where it is made available and use English for the header in order to make it usable.

7. PLOS authors have the option to publish the peer review history of their article (what does this mean?). If published, this will include your full peer review and any attached files.

Reviewer #1: No

Reviewer #2: No

Reviewer #3: No

---

## [Author Response · Author response to Decision Letter 1]

5 Jan 2024

Response to Reviewer 2 Comments

Dear reviewer 2:

Thank you very much for your time involved in reviewing the manuscript, and your comments have further improved the quality of the manuscript.

We have carefully reviewed the comments and revised the manuscript accordingly. The modified section was already highlighted in yellow. Hope the explanation has fully addressed all of your concerns. Point-by-point response to reviewer are attached below this letter.

Please see the attachment.

Q1. State the research question clearly in the Introduction.

Reply: Many thanks for your comments. We have added the research question of this study in the Introduction, for specific modifications see lines 66-72 of the manuscript.

Q2. List the main contributions in the Introduction.

Reply: Thank you for your suggestions. We have added the main contributions in the Introduction, for specific modifications see lines 72-75 of the manuscript.

Q3. Provide paper structure at the end of the Introduction.

Reply: Many thanks for your comments. We have provided paper structure, for specific modifications see lines 174-193 of the manuscript.

Q4. Literature survey should be expanded, especially with recent papers dealing with COVID19. You can include the following:

Reply: Thank you for the professional references you provided. We have carefully read these references and added them to the manuscript. For the specific content, please see lines 42-46, 347-372 and 373-393 of the manuscript.

Q5. The contribution of the research to the field should be justified better.

Reply: Thank you for your suggestions. We have added the contribution of the research to the field, for specific modifications see lines 373-393 of the manuscript.

Q6. Provide more details about the utilized questionary.

Reply: Many thanks for your comments. We have provided more details about the utilized questionary. For specific content see Table 3 (lines 283-286) .

Q7. Discussion should be more elaborate.

Reply: Thank you for your suggestions. We have enriched the discussion, for specific modifications see lines 347-372 of the manuscript.

Q8. Discuss the limitations of the research as well.

Reply: Many thanks for your comments. The discussion of the limitations and the future research recommendations are presented in lines 429-439 of the manuscript.

Q9. Conclusion is very limited, it should summarize up the research, mention the most important findings and limitations of the research, and indicate future research in this topic.

Reply: Thank you for your suggestions. We have enriched the Conclusion, for specific modifications see lines 416-422 of the manuscript. Furthermore, the discussion of the limitations and the future research recommendations are presented in lines 429-439 of the manuscript.

Q10. Thorough proofreading of the paper is recommended.

Reply: Thank you for your suggestions. We have proofread the revised manuscript.

Response to Reviewer 3 Comments

Dear reviewer 3:

Thank you very much for your time involved in reviewing the manuscript and your comments have further improved the quality of the manuscript.

We have carefully reviewed the comments and revised the manuscript accordingly. The modified section was already highlighted in yellow. Hope the explanation has fully addressed all of your concerns. Point-by-point response to reviewer are attached below this letter.

Please see the attachment.

Point: The data set that is made available is not intelligible. The columns of the table are not in English and the meaning of the values is not explained. Please add a description of the data set where it is made available and use English for the header in order to make it usable.

Response: Many thanks for your comments. We have unified the abbreviations in the data with those within the manuscript and changed them to English. Meanwhile, we also submitted a "Data availability statement" with this statement as a detailed description of the data set and the meaning of these abbreviations.

The data used in this paper and the description of the data set are openly available at the OSF site for this paper at https://osf.io/fme5s/, DOI 10.17605/OSF.IO/FME5S.

---

## [Decision Letter · Decision Letter 2]

17 Jan 2024

Intolerance of Uncertainty and Mental Health in China “Post-pandemic” Age: The Mediating Role of Difficulties in Emotion Regulation

PONE-D-23-13232R2

Dear Dr. Li,

We’re pleased to inform you that your manuscript has been judged scientifically suitable for publication and will be formally accepted for publication once it meets all outstanding technical requirements.

Kind regards,

Nebojsa Bacanin

Academic Editor

PLOS ONE

Additional Editor Comments (optional):

Dear Authors,

thank you for revising your manuscript.

Reviewers' comments:

Reviewer's Responses to Questions

**Comments to the Author**

1. If the authors have adequately addressed your comments raised in a previous round of review and you feel that this manuscript is now acceptable for publication, you may indicate that here to bypass the “Comments to the Author” section, enter your conflict of interest statement in the “Confidential to Editor” section, and submit your "Accept" recommendation.

Reviewer #2: All comments have been addressed

Reviewer #3: All comments have been addressed

2. Is the manuscript technically sound, and do the data support the conclusions?

Reviewer #2: (No Response)

Reviewer #3: (No Response)

3. Has the statistical analysis been performed appropriately and rigorously? 

Reviewer #2: (No Response)

Reviewer #3: (No Response)

4. Have the authors made all data underlying the findings in their manuscript fully available?

Reviewer #2: (No Response)

Reviewer #3: (No Response)

5. Is the manuscript presented in an intelligible fashion and written in standard English?

Reviewer #2: (No Response)

Reviewer #3: (No Response)

6. Review Comments to the Author

Reviewer #2: (No Response)

Reviewer #3: (No Response)

7. PLOS authors have the option to publish the peer review history of their article (what does this mean?). If published, this will include your full peer review and any attached files.

Reviewer #2: No

Reviewer #3: No

---

## [Editor Report · Acceptance letter]

23 Jan 2024

PONE-D-23-13232R2 

PLOS ONE

Dear Dr. Li, 

I'm pleased to inform you that your manuscript has been deemed suitable for publication in PLOS ONE. Congratulations! Your manuscript is now being handed over to our production team.

Kind regards, 

on behalf of

Dr. Nebojsa Bacanin 

Academic Editor

PLOS ONE